# Transformers for Energy Forecast

**DOI:** 10.3390/s23156840

**Published:** 2023-08-01

**Authors:** Hugo S. Oliveira, Helder P. Oliveira

**Affiliations:** 1Institute for Systems and Computer Engineering, Technology and Science—INESC TEC, University of Porto, 4200-465 Porto, Portugal; holiveira@fc.up.pt; 2Faculty of Sciences (FCUP), University of Porto, 4169-007 Porto, Portugal

**Keywords:** transformers, time-series forecast

## Abstract

Forecasting energy consumption models allow for improvements in building performance and reduce energy consumption. Energy efficiency has become a pressing concern in recent years due to the increasing energy demand and concerns over climate change. This paper addresses the energy consumption forecast as a crucial ingredient in the technology to optimize building system operations and identifies energy efficiency upgrades. The work proposes a modified multi-head transformer model focused on multi-variable time series through a learnable weighting feature attention matrix to combine all input variables and forecast building energy consumption properly. The proposed multivariate transformer-based model is compared with two other recurrent neural network models, showing a robust performance while exhibiting a lower mean absolute percentage error. Overall, this paper highlights the superior performance of the modified transformer-based model for the energy consumption forecast in a multivariate step, allowing it to be incorporated in future forecasting tasks, allowing for the tracing of future energy consumption scenarios according to the current building usage, playing a significant role in creating a more sustainable and energy-efficient building usage.

## 1. Introduction

Building energy efficiency has become increasingly important as climate change and energy security concerns have grown [1]. Building energy usage accounts for significant global energy consumption and greenhouse gas emissions. Improving building energy efficiency is a crucial strategy for reducing energy consumption and mitigating climate change.

One emerging technology that can potentially improve building energy efficiency is energy forecast models, trained from live data from sensors and other sources [2].

Energy forecasts can be used to improve building energy efficiency in several ways. One approach is to optimize the operation of building systems, such as heating, ventilation, and air conditioning (HVAC) systems [3] according to the current demand.

By forecasting and analyzing the building’s energy performance, operators can identify ways to reduce energy consumption while still maintaining comfort and safety. Another approach is to use the forecast models to identify and prioritize energy efficiency upgrades [4].

Developing an effective energy forecast model requires heterogeneous sensor data to unveil hidden building usage patterns. The Figure 1 diagram shows how building sensor data combined with additional data and machine-learning models allow all available data to represent the building energy consumption pattern.

In addition to improving the building energy efficiency, forecasting energy consumption allows operators to identify ways to optimize the structure for different use cases by simulating different scenarios.

The energy forecast model is based on modifying the multi-attention transformer model by including a learnable weighting feature attention matrix to address the building energy efficiency. The model is leveraged by analyzing live sensor data from the Institute for Systems and Computer Engineering, Technology and Science (INESC TEC) Research Building, consisting of computer research labs, auditoriums and support facilities, alternating more than 700 researchers, staff and other personnel.

The objective is to forecast the energy consumption for the next 10 days, allowing the application of effective mitigation procedures regarding energy waste and quickly assessing how these mitigation procedures impact the actual energy consumption.

With this in mind, this research work is organized into sections, with Section 2 presenting a comprehensive study on forecast models works, focused on energy forecast and maintenance; Section 3 showing the principal methodology to be followed and techniques to be employed; Section 4 containing the data analysis and forecast modeling; Section 5 containing the results discussion; and the conclusion in Section 6.

## 2. Literature Review

Building energy forecast models are an excellent tool for understanding and mitigating infrastructure efficiency in many fields. They allow for the creation of several data scenarios to foresee the impact on energy consumption and evaluate the impact of building usage modifications.

Maintenance decision-making and sustainable energy are the main concerns of [5], by employing the forecasting of electrical energy consumption in equipment maintenance by means of an artificial neural network (ANN) and particle swarm optimization (PSO). With the same objective, [6] employs fully data-driven analysis and modeling by first analyzing the linear correlation between the outdoor environmental parameters with the actual measured energy consumption data and then employing the use of the backpropagation artificial neural network (BP-ANN) to forecast energy consumption, allowing the reduction of the low-carbon operation and maintenance for the building HVAC systems.

The issue of CO_2_ emissions poses a significant challenge to building efficiency. In the study conducted by [7], the research delves into both the advantages and limitations of conventional energy consumption mitigation methods. Notably, the study emphasizes the potential of integrating 2D GIS and 3DGIS (CityGML) [8] with energy prediction approaches. This combination considers frequent interventions at the building scale, applicability throughout the building’s life cycle and the conventional energy consumption forecasting process.

In the work of [9], the emphasis is placed on the significance of forecasting energy usage in building energy planning, management and optimization. The review acknowledges the potential of deep learning approaches in effectively handling vast quantities of data, extracting relevant features and enhancing modeling capabilities for nonlinear phenomena.

The use of LSTM and GRU for energy forecasting was the subject of study for [10]. LSTM combinations with CNN were proposed by [11] for the same purpose. The CNN-LSTM proposed model employs a CNN to extract complex non-linear features combined with a LSTM to handle long-term dependencies through modeling temporal information in the time-series data.

LSTM, Bi-LSTM and GRU are employed in [12] to perform occupancy prediction in buildings with different space types. Another example related to intelligent vehicle systems can be found in [13], where the authors evaluated the performance of different architectures, including LSTM, to predict vehicle stop activity based on vehicular time-series data.

In a similar way, [14] exploits CNN–LSTM models for household energy consumption, with a particular emphasis on data normalization. The transformers for time series are also a subject of interest in [15], proposing Autoformer, a novel architecture with an auto-correlation mechanism based on the series periodicity, which conducts the dependencies discovery and representation aggregation at the sub-series level, outperforming self-attention in both efficiency and accuracy.

Although energy forecasting models mostly target foreseen target variables, they can be employed to define maintenance operations, allowing for the mitigation of the operational cost in building or manufacturing facilities [16].

The current time-series forecast models have several limitations, namely, in exploring all the available information in a meaningful way, harming the forecasting of a longer period of time. This concern makes it vital for developing models that use all information in a multivariate way and captures very long patterns useful in a correct energy consumption forecast.

## 3. Methodology

The main objective of this work is to employ a modified multi-variable transformer to forecast energy building consumption for the next 10-day period (250 h).

To establish a comparison baseline, multistep LSTM/GRU models are also employed and trained with the same data. The overall description of the proposed and evaluated models is below.

### 3.1. Baseline Models

lstm [17] is a type of rnn suitable for time-series forecasting. It is designed to address the vanishing gradient problem in traditional rnns that occurs when the gradients used to update the network weights become very small when propagating through many time steps, harming the learning of long-term dependencies.

The lstm models consist of a series of lstm cells (Figure 2), with the number of cells as a hyperparameter. Each of these cells contains three gates (input, forget and output) that control the flow of information through the cells. The weight of these gates is learned during the training process. Each cell stores information over time, and a hidden state allows it to pass information between cell chains in the network.

The equations for the long short-term memory (LSTM) model are as follows:$$\begin{array}{cc} \mathrm{Input}\;\mathrm{Gate}:\; & i_{t}=\sigma \left(W_{xi}x_{t}+W_{hi}h_{t-1}+W_{ci}c_{t-1}+b_{i}\right)\\ \mathrm{Forget}\;\mathrm{Gate}:\; & f_{t}=\sigma \left(W_{xf}x_{t}+W_{hf}h_{t-1}+W_{cf}c_{t-1}+b_{f}\right)\\ \mathrm{Cell}\;\mathrm{State}\;\mathrm{Update}:\; & {\tilde{c}}_{t}=\tanh\left(W_{xc}x_{t}+W_{hc}h_{t-1}+b_{c}\right)\\ \mathrm{Cell}\;\mathrm{State}:\; & c_{t}=f_{t}⊙c_{t-1}+i_{t}⊙{\tilde{c}}_{t}\\ \mathrm{Output}\;\mathrm{Gate}:\; & o_{t}=\sigma \left(W_{xo}x_{t}+W_{ho}h_{t-1}+W_{co}c_{t}+b_{o}\right)\\ \mathrm{Hidden}\;\mathrm{State}:\; & h_{t}=o_{t}⊙\tanh\left(c_{t}\right) \end{array}$$
where
$i_{t}$ represents the input gate activation at time step *t*,$f_{t}$ represents the forget gate activation at time step *t*,${\tilde{c}}_{t}$ represents the candidate cell state at time step *t*,$c_{t}$ represents the cell state at time step *t*,$o_{t}$ represents the output gate activation at time step t,$h_{t}$ represents the hidden state (output) at time step *t*,$x_{t}$ represents the input at time step *t*,$h_{t-1}$ represents the hidden state at the previous time step (t−1),$c_{t-1}$ represents the cell state at the previous time step (t−1),σ represents the sigmoid activation function,⊙ represents the element-wise multiplication (Hadamard product).

These equations describe the operations performed by an LSTM cell to update and pass information through time steps in a recurrent neural network architecture.

The memory cell is updated based on the input, forget, and output gates, while the hidden state is updated based on the memory cell and the output gate. In summary, LSTM has these particular hyperparameters: the number of LSTM layers that determine the network depth; the number of LSTM units, allowing for the definition of the size of the hidden state and memory cell, controlling the network ability to store information.

GRU [18] is also a type of RNN widely used in time-series forecasting. It addresses the same problem as lstm through a simpler architecture, described in Figure 3.

The GRU cell also contains a hidden state, which is used to pass information between cells in the network. Unlike LSTM, GRU only has one memory cell, which is updated using the reset and update gates. Both models commonly employ a Dropout technique [19] to prevent overfitting and adaptable learning rate usage, such as Adam [20] or SGD [21].

The equations for the gated recurrent unit (GRU) model are as follows:


**Update Gate:**

$$\begin{array}{cc} z_{t} & =\sigma (W_{xz}x_{t}+W_{hz}h_{t-1}+b_{z}) \end{array}$$




**Reset Gate:**

$$\begin{array}{cc} r_{t} & =\sigma (W_{xr}x_{t}+W_{hr}h_{t-1}+b_{r}) \end{array}$$




**Candidate Hidden State:**

$$\begin{array}{cc} {\tilde{h}}_{t} & =\tanh(W_{xh}x_{t}+W_{hh}\left(r_{t}⊙h_{t-1}\right)+b_{h}) \end{array}$$



**Hidden State Update:**$$\begin{array}{cc} h_{t} & =\left(1-z_{t}\right)⊙h_{t-1}+z_{t}⊙{\tilde{h}}_{t} \end{array}$$ where
$z_{t}$ represents the update gate activation at time step *t*,$r_{t}$ represents the reset gate activation at time step *t*,${\tilde{h}}_{t}$ represents the candidate hidden state at time step *t*,$h_{t}$ represents the hidden state (output) at time step *t*,$x_{t}$ represents the input at time step *t*,$h_{t-1}$ represents the hidden state at the previous time step (t−1),*W* represents weight matrices,*b* represents bias vectors,σ represents the sigmoid activation function,⊙ represents the element-wise multiplication (Hadamard product).

These equations describe the operations performed by a GRU cell to update and pass information through time steps in a recurrent neural network architecture.

Overall, LSTM and GRU architectures are both effective for time-series forecasting, and the choice between the two will depend on the specific requirements of the task at hand. In general, LSTM is a better choice for tasks requiring capturing long-term dependencies, while GRU may be more appropriate for tasks requiring faster training times or datasets with shorter-term dependencies. Although many variations of LSTM and GRU were proposed to improve forecasting capabilities, such as a bidirectional LSTM [22], by utilizing information from both sides, or a bidirectional GRU [23] with the same purposes, the main fundamental architecture is used.

### 3.2. Proposed Transformer Multistep

The transformers model [24] employs the use of an encoder–decoder scheme formed by a set of stacked self-attention in combination with point-wise layers to map the input sequence $\left(x_{i},\cdots ,x_{n}\right)$ to a series of continuous representations $z=(z_{1},\cdots ,z_{n})$. For a given *z*, it generates a set of output symbols $\left(y_{i},\cdots ,y_{m}\right)$ at each time, in an auto-regressive manner, using the previously generated forecast points as additional inputs.

The sequence data points are transformed into discrete tokens, converted into a numeric token representation and fed into the input embedding layer to map the sequence element into a continuous learnable vector.

Because the proposed transformer model does not contain any recurrence or convolution blocks to insert information regarding the relative position of the input tokens among the input sequence, a piece of positional encoding information is inserted into the embedding layer that corresponds to a cosine and sine relative function representation, generating two separate vectors from the even $p_{e}\left(m,2n\right)$ and odd $p_{e}\left(m,2n+1\right)$ sequence time steps, embedding the positional information based on https://github.com/oliverguhr/transformer-time-series-prediction, accessed on 30 May 2023).

The embedding vector is defined by the dimension *D* and the positional embedding $p_{e}\left(m,2n\right)$ with the even elements of the positional vector *P* for a given input *X* in each time step represented in $p_{e}\left(m,2n\right)$, resulting in Equation (Equation 1):(1)$$p_{e}\left(m,2n\right)=\sin m^{\left[-2n\;\log(1000)/D\right]}$$

Regarding the odd element representation, each positional embedding $p_{e}\left(m,2n+1\right)$ is expressed in Equation (Equation 2).
(2)$$p_{e}\left(m,2n+1\right)=\cos m^{\left[-2n\;\log(1000)/D\right]}$$

The final positional encoded *P* vector aggregates both the even and pairs positional encoding, resulting in a final embedding vector with X+P dimensions.

Concerning the encoding, layers are formed by a stack of $N_{l}$. This hyperparameter corresponds to the number of stacked identical layers. Each layer is formed by a multi-head self-attention mechanism combined with the positional-wise fully connected feed-forward network. Each sub-layer employs a residual connection followed by a layer normalization, with the output expressed in Equation (Equation 3).
(3)Output=LayerNorm(y)+SubLayer(y))

All sub-layers and embedding layers inside the model generate an output dimension *D* equal to model size $d_{model}$ considering the input features inputfeature. In this case, the optimal value was found to be 250×inputfeature to encompass the full 10-day forecast period.

The decoder is very similar to the encoder. However, it adds a third sub-layer to perform multi-head attention over the output of the stacked encoders and two sub-layers, using similar residual connections as the encoder and adjacent layer normalization. Similarly, the number of decoder layers is a hyperparameter to be determined. The final decoder stack embedding output is compensated by one position, allowing the avoidance of current positions being mixed with subsequent ones when using squared subsequent masking.

The input queries *Q*, keys *K* and values *V* of dimension $d_{v}$ are subject to the scaled dot product of the keys with all given keys, normalized by the dimension of keys $d_{k}$ as $\sqrt{d_{k}}$, to overcome the monotonous magnitude growth with the increase of the *d* dimension, leading to gradient vanish (Figure 4). The final attention matrix *A* that encapsulates the packed Q,K,V is obtained using a softmax to gather the weights of the values, being represented in Equation (Equation 4)
(4)$$A\left(Q,K,V\right)=softmax\left(\frac{QK^{T}}{\sqrt{d_{k}}}\right)V$$

Figure 4 shows how Queries *Q*, Keys *K* and Values *V* are combined in an incremental dot product, passing by the scaling layer, optional masking layer, softmax and final matrix multiplication to form the attention matrix *A*.

The train of transformers allows us to effectively construct an attention matrix formed by queries *Q*, keys *K* and value *V*.

Transformers models can contain single or multi-attention heads, with a single head putting all focus in a single location, aggregating all contributions to a location with the same weight. The main drawback of single-attention heads is that they lead to averaging the contributions to a local representation.

Alternatively, multi-head attention (Figure 5 allows the modeling of several representations from different locations simultaneously, allowing the capture of information from several sparse locations with different weights, expressed in Equation (Equation 5). For each of the projected queries $d_{k}$, $d_{k}$ and $d_{v}$, attention is performed in parallel, resulting in $d_{v}$ dimensional value representations, and the final heads *h* becoming concatenated into a single attention output.

The final Equation (Equation 5) for multi-head attention can be expressed as
(5)$$MultiHead\left(Q,K,V\right)=Concat\left(h_{1},h_{2},\cdots ,h_{n}\right)W^{O}$$
with $h_{n}$ being the number of attention heads and *n* being the projections matrices, with each head represented as $head_{i}=Attention\left(QW_{i}^{Q},KW_{i}^{K},V_{i}^{W}\right)$, containing *Q* queries and *K* keys as a result of the dot product, and the projections matrices that correspond to parameters matrices $W_{i}^{Q}\in R^{D\times d_{k}},W_{i}^{K}\in R^{D\times d_{k}}$ and $W_{i}^{V}\in R^{D\times d_{v}}$. Following this operation, the outputs are concatenated and multiplied by the weighting matrix $W^{O}$, corresponding to a squared matrix obtained from $R^{hdv}\times D$ [24].

The proposed multistep transformers with the modified attention heads and input embedding follow the diagram of Figure 6, which is largely based on [24]. However, with a modification on the multi-attention heads and how the inputs *X* are combined, we use a correspondent embedding through a learnable weight attention matrix instead of a dedicated embedding for each input feature. The final proposed input aggregated embedding is expressed in Equation (Equation 6).
(6)$$\epsilon_{t}=\sum_{j=1}^{n}m_{X_{t}}^{\left(j\right)}\epsilon_{t}^{(j}$$
where ${\hat{\epsilon }}_{t}$ corresponds to the linear transformer feature input, $m_{X_{t}}$ is the leaned matrix weight regarding the combination of each of the input transformed features at time *t* towards the final embedding ${\hat{\epsilon }}_{t}$ at instant *t*.

The proposed multistep multi-head transformer model is represented in Figure 6, with inputs $X_{1}$ to $X_{n}$ at instant m−k, corresponding to the different input features at a particular period of time, combined in an attention head $A_{{}_{m}}$ from Equation (Equation 6) for each considered instant, following by the aggregated embedding and the combination of the positional encoding and transformer block to form the full encoder stage. The decoder stage employs the transformer decoder block, the forecast layer and the dense for each target future value.

## 4. Setup And Forecasting

To evaluate the performance of the proposed multivariate transformer, two comparison baselines, LSTM and GRU, were trained on the same data with the same objective. For easy modeling, the redundant sensors are averaged into a single one per room to construct the feature regarding each room measurement.

### 4.1. Models

To evaluate the performance of the proposed transformer and baseline models, after training, the models infer over 250 points of historical data (test input that corresponds to 10 days) and forecast another 250 hourly future points (corresponds to the 10 day period ahead). For training, 70% of the data starting from day 1 are set for training, the subsequent 10% chunks for validation and 20% for testing purposes.

In more detail, the proposed transformer blocks follow the arrangement of Figure 6 with an aggregation of the multivariate input features using Equation (Equation 6).

The LSTM and GRU models use 250 units cells to forecast the same period length, described in Figure 7.

Considering the multistep forecast, an additional repeat vector layer and time-distributed dense layer are added to the architecture.

### 4.2. Evaluation Metrics

The trained models were evaluated using MSE and MAPE on each time period $\Delta_{t}$. The forecasted value $\hat{y_{i}}$ subtracted from the actual one $y_{i}$ and divided by $y_{i}$ and normalized by the set of samples *N*, is expressed as
(7)$$\mathrm{MAPE}\left(y,\hat{y}\right)=\frac{1}{N}\sum_{i=1}^{N}\left|\frac{y_{i}-\hat{y_{y}}}{y_{i}}\right|.$$

MSE considers the average of the squared errors between the real value $y_{i}$ and the predicted $\hat{y_{i}}$ in a given *N* set of samples, expressed as
(8)$$\mathrm{MSE}\left(y,\hat{y}\right)=\frac{1}{N}\sum_{i=1}^{N}{\left(y_{i}-\hat{y_{i}}\right)}^{2}.$$

### 4.3. Dataset

To construct a model capable of forecasting energy consumption, the first task is to analyze the data produced by several sensors during one year. The activities conducted at the building are mainly research regarding computer science and electrical engineering and support services, composed of research labs, human resources, auditoriums, restaurants and other common building infrastructure. The building comprises two blocks of four floors encompassing several room characteristics, such as computer labs, meeting auditoriums, restaurants and service spaces.

Each of the individual rooms contains a set of sensors to measure environmental variables and energy consumption through a dedicated sensor network. The deployed sensors on each room perform a combined measurement of **humidity**; **temperature**; **Co2** concentration, **pressure**, expressed in Pascals; and local energy consumption, for a total of 144 sensors.

The dataset includes the following:Real-time historical data from the INESC TEC building.Encompassing two-year time span.Totaling 8760 × 2 sample points (2 years).

### 4.4. Data Analysis

Regarding the data in analysis, the mentioned dataset comprises the building rooms’ ambient sensory and corresponding energy consumption gathered during a year. The variables in the dataset contain the interior temperature in degrees Celsius ${}^{\circ }$C, relative humidity in percentage %, air pressure expressed in Pascals PPa and room energy consumption as a target variable gathered in 15-min intervals throughout the year, aggregated in hour means. The analysis is restricted to only one year of data for a more concise analysis.

Concerning the energy consumption, Figure 8 presents the average energy consumption per room considering the day period and weekday.

Figure 8 clearly highlights that weekends account for lower consumption, and the morning and noon periods account for a large part of the total building consumption, corroborated by the fact that it is a computer science research lab. However, it is clear that Thursday accounts for a peak in average room consumption, with Friday showing a clear descent in energy consumption.

Regarding the energy consumption among the building’s main floors and sections, Figure 9 aggregates the average consumption per floor.

From Figure 9, it is possible to identify a clear outlier, namely, floor B0, which corresponds to the location of the restaurant and cafeteria of the building, where large energy kitchen appliances are present. In the opposite direction, floor B5 accounts for the lowest average energy consumption, correlated by the fact that this floor only contains small building auxiliary devices. The restaurant information can be corroborated by Figure 10.

Figure 10 exhibits that floor B0 has a higher energy peak in the morning and at noon compared to the second largest floor consumption (A4), corresponding to the services floor. The consumption on floor B0 intersects with the most agitated period of the restaurant at lunchtime.

Figure 11 shows the mean building consumption by month, with clear evidence that the August months account for a lower consumption due to vacations and the winter months for a substantial increase in consumption due to lighting and heating.

Regarding environment variable sensors, Figure 12 shows the box-plot distribution of the ambient variables by room.

From Figure 12 there is clear evidence that auditoriums present higher humidity levels due to less usage and exposure to the sun. Considering these large auditoriums are located on the lower floor, they present with higher humidity due to a lower temperature (Figure 13), a well-known ambient variable phenomenon.

### 4.5. Time-Series Analysis and Pre-Processing

To accurately forecast energy consumption, it is relevant to determine the presence of seasonality, trends and the presence of abnormal values.

ACF corresponds to the correlation between a time series with a lagged version of itself, up to 50 lags, starting at a lag of 0 and having the maximum correlation at this period of time. It allows us to determine if a time series corresponds to white noise/random, the degree of the relation of a given observation regarding its adjacent observation and determine the order of the time series. Additionally, PACF allows the inclusion or exclusion of indirect correlations in the ACF calculation. Figure 14 summarizes the main time-series correlation (considered only 10 lags). The blue area depicts the 95% confidence interval, meaning that anything within the blue area is statistically close to zero, and anything outside the blue area is statistically non-zero with regard to the target variable, **EnergyWh**, the ACF and PACF in Figure 14, and derived features such as weekday (Monday to Sunday), day period (‘Late Night’, ‘Early Morning’, ‘Morning’, ‘Noon’, ‘Eve’, ‘Night’), and season of the year (spring, summer, autumn, winter).

Shows that for most lags, the auto-correlation is significantly non-zero for all lags. Therefore, the time series is not random, presenting some degree of seasonal patterns.

In order for models to converge properly, the dataset is pre-processed to find outliers or erroneous measures that may harm the training of the models, replacing those values with the mean of the previous and next values $X_{i}=\frac{1}{2}\left(x_{i-1}+x_{i+1}\right)$. Furthermore, the data were normalized with respect to min–max to be fed to models, and the reverse process was made to recover the target forecasted value.

## 5. Results

Several combinations of hyperparameters were used to train and evaluate the three models effectively. All models were trained during 100 epochs in a 2 × Intel Xeon Gold 2.5 Ghz with 20 cores each, totaling 384 GB RAM, fitted with 2 Teslas v100 with 32 Gb each and 2 GTX 2080 with 11 Gb each. A 5 k-CV was employed to determine the set of the best hyperparameters for each configuration and to evaluate the model in the test set.

In order to compare different time-series forecasts, MAPE was employed, since it is a scale-independent measurement, and our time series does not cross zero, so the undefined problem is circumvented.

The final transformer model architecture took approximately 20 h of training, with an average of 1000 s on each epoch, approximately 1.6 times higher than LSTM and 1.9 than GRU, with the final convergence curves represented in Figure 15.

From Figure 15, it is evident that GRU models presented some degree of overfitting, compared with LSTM and the transformer proposed model. The transformer model shows a more stable convergence during the set of epochs, mainly due to a larger complexity when compared with other RNN models.

After curated debugging and hyperparameter tuning, the transformer with six heads stacked with six identical encoders obtained the best result (Table 1), with a dropout max of dropout=0.2.

The decoder block employs the use of a linear transformation of the input data into the same number of $in_{features}=features_{size}=250\times n$ and $out_{feature}=1$, corresponding to the forecast of 10 days (250 h).

The loss was the L2-norm, employing SGD with a gamma=0.98, a learning rate of lr=0.005 and $step_{s}ize=10$ using a batch the size of 16. The transformer’s total number of parameters was 16,330,827, with a look-back window of 34.

Regarding LSTM, the optimal number of nodes is 256 nodes per layer (Table 1), using a dropout of 0.2, a decay rate 0.99, with a look-back window of 32, employing Adam optimization with a learning rate/step size of lr=0.005, with $\beta_{1}=0.9$ and $\beta_{2}=0.99$, using a L2-loss with a batch size of 16.

Regarding GRU, the optimal number of nodes is 200, 128 nodes per layer, a dropout of 0.2, Xavier weight initialization, decay rate 0.99, with a look-back window of 26, employing Adam optimization with a learning rate/step size of lr=0.005, with $\beta_{1}=0.9$ and $\beta_{2}=0.99$, using a L2-loss with a batch size of 16.

Table 1 summarizes the results obtained in all the evaluated models using the test input samples (MAPE and MSE).

Figure 16 presents a side-to-side comparison of the top performer models against the ground-true values in the test set.

Choosing an input window is crucial for achieving optimal performance. It is important to analyze the input data to identify any seasonal patterns thoroughly, and the input sequence size should include these personalities or trends. In this case study, a look-back window of 32 to forecast 250 points ahead with six heads, both in the encoder and the decoder, resulted in the best overall performance on the proposed transformer model. In the transformer models, an input window lower than 250 time steps led to suboptimal results. This corresponds to a suboptimal selection of input-target sequences that do not capture relevant time-series patterns useful for the model to forecast future energy values correctly.

This study aimed to assess the performance of multi-head attention-based transformers for the energy consumption forecast and compare the performance with LSTM and GRU-based models.

Transformers have proven to be highly suitable for multivariate time-series forecasting using large data samples and the correct use of the input window. The multi-head attention mechanism increases the performance, particularly in tasks involving multi-step forecasting. Implementing such models is widely applicable across various sectors, including energy forecasting, where they can aid in establishing mitigation policies to reduce operational costs and address challenges related to climate management. Additionally, they can be beneficial for modeling building maintenance and predictive control for HVAC systems [16,25], bringing operational benefits.

However, further changes and improvements are necessary to enhance the model’s efficiency and robustness to ensure greater competence in real-world data-driven models for different sectors.

## 6. Conclusions

Energy forecasting is crucial for building characterization, which often manifest unpredictable energy consumption patterns that are not captured by models, leading to a degradation of their performance. Deep learning approaches, on the contrary, allow for a model of non-linear dependencies and capture relevant information to perform forecasts.

In this article, a modification of a multi-head multivariable transformer model is proposed for building an energy consumption forecast, complemented by a comprehensive performance comparison with common RNN models, such as GRU and LSTM and a real-time building energy and environment collected dataset. MSE and MAPE were used to evaluate the models. The performance of the multivariate transformer model using a multi-head attention mechanism and modified input embedding is almost 3.2 p.p. better than the best-trained baselines.

The construction of a richer dataset must follow the guidelines of the ROBOD project [26], which encompasses many sensors, HVAC, building occupancy and WiFi traffic, among others.

The main drawback of the multi-head attention transformer concerns its complexity and training time and the limited set of training features. Thus, training in a more rich dataset such as the Building Data Genome Project 2 [27] or the Global Occupant Behaviour database [28] would enable us to further validate the proposed forecast model with more variables, allowing us to foresee potentialities in the use of transformers in the current energy forecast, providing accurate energy consumption forecasts.

## Figures and Tables

**Figure 1 sensors-23-06840-f001:**
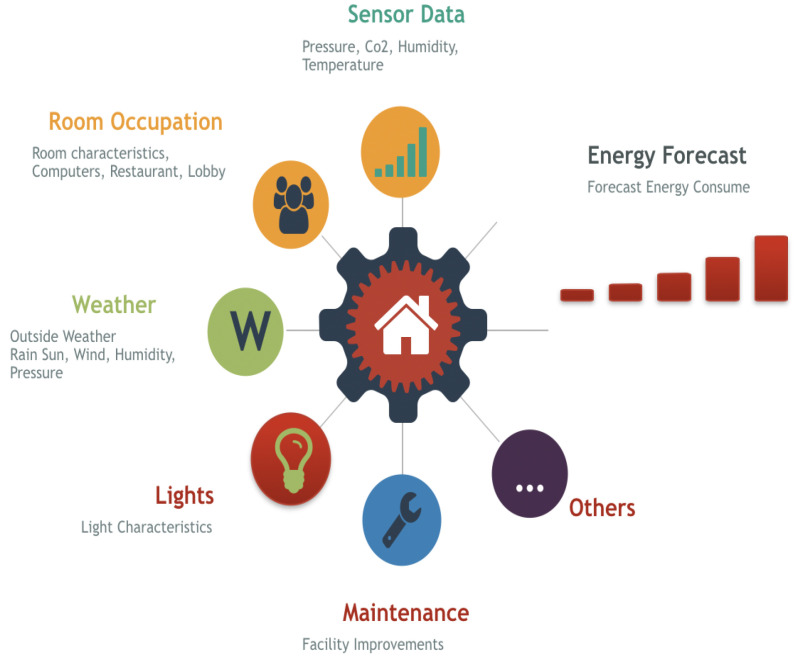
Energy Consumption Forecast.

**Figure 2 sensors-23-06840-f002:**
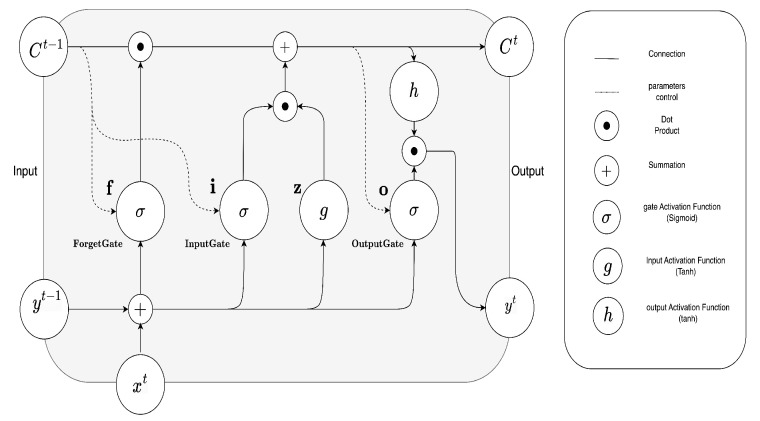
LSTM cell block.

**Figure 3 sensors-23-06840-f003:**
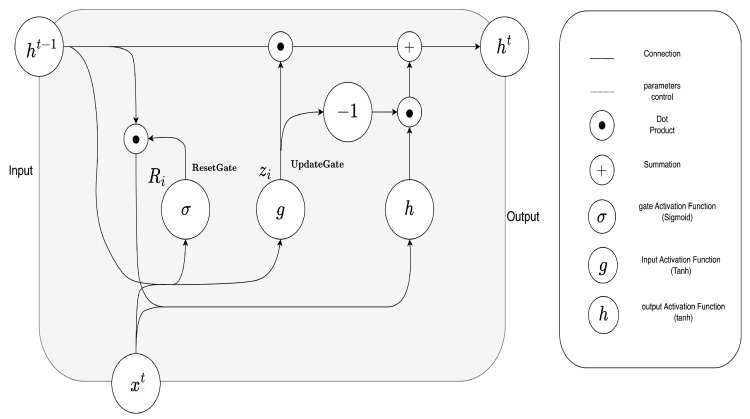
GRU cell block.

**Figure 4 sensors-23-06840-f004:**
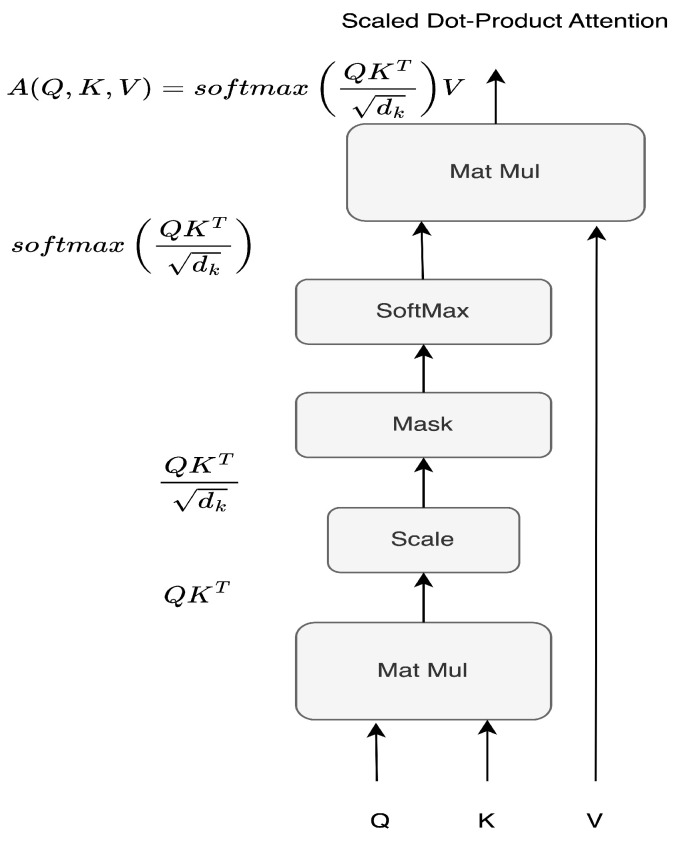
Scaled dot product attention scheme.

**Figure 5 sensors-23-06840-f005:**
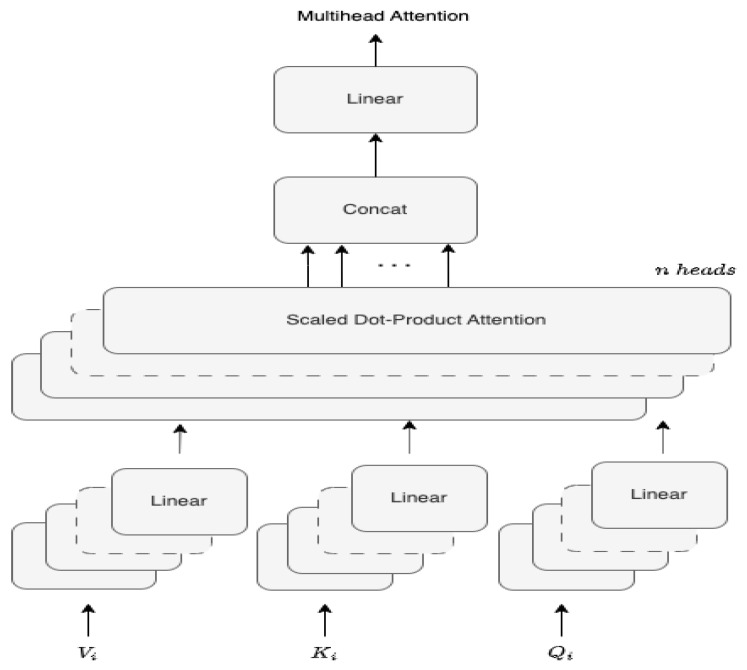
Multihead attention scheme.

**Figure 6 sensors-23-06840-f006:**
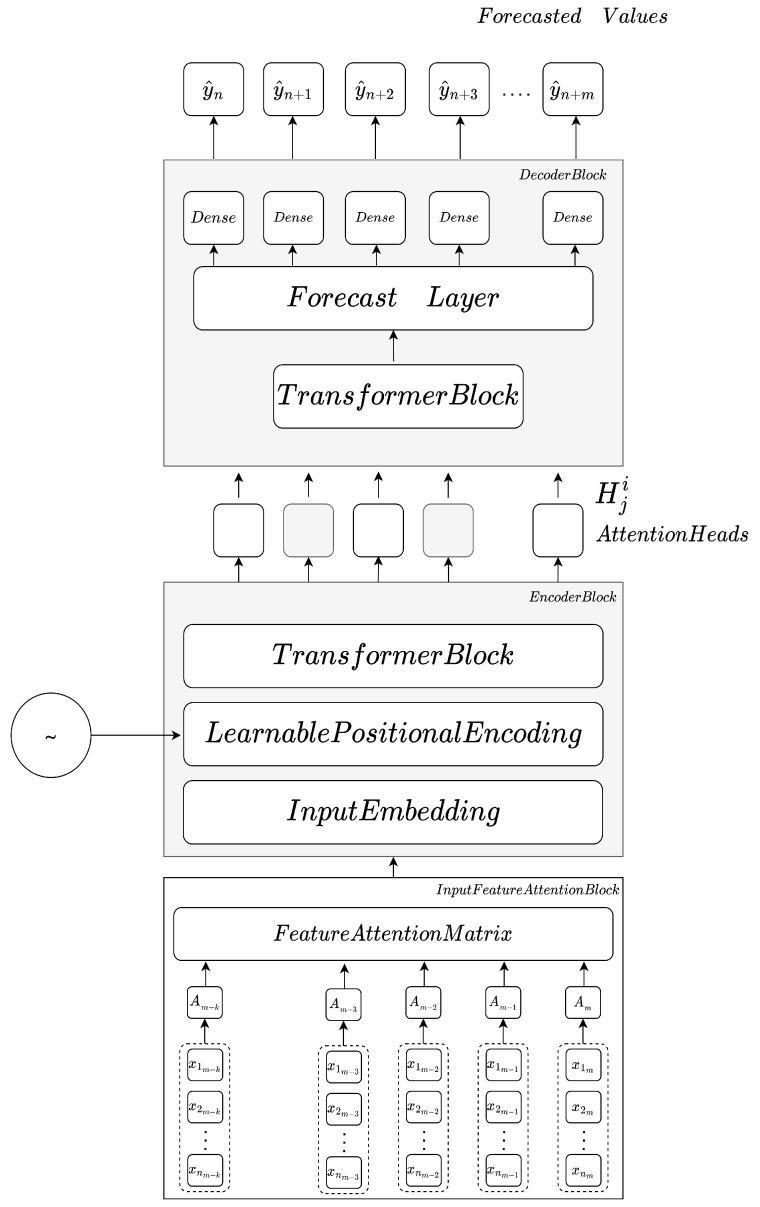
Transformer Proposed Model.

**Figure 7 sensors-23-06840-f007:**
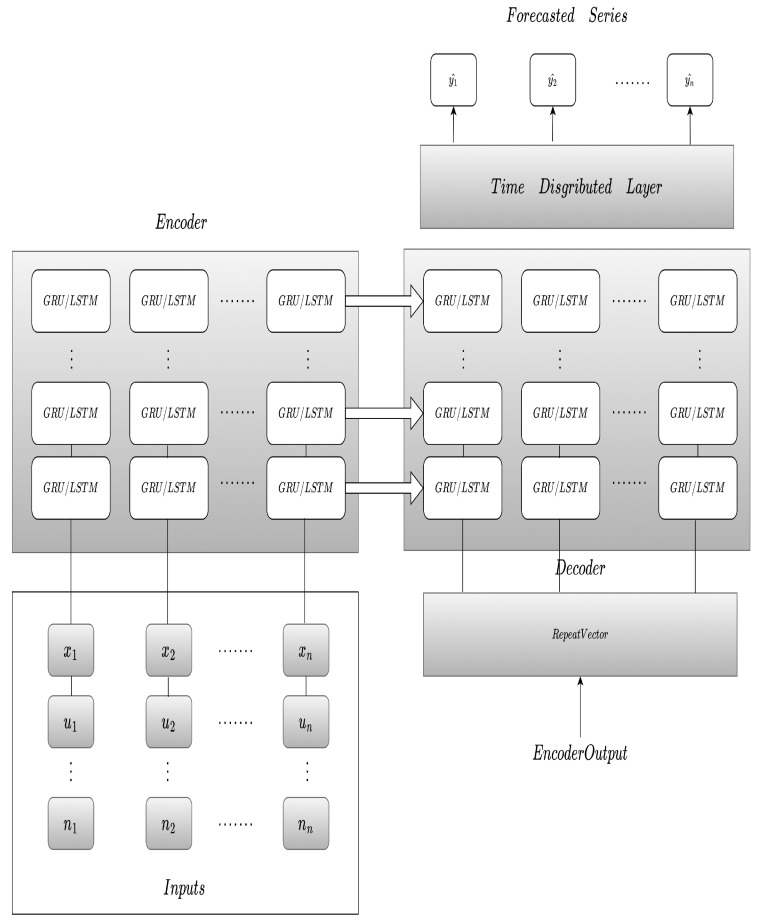
LSTM and GRU time-series forecast model.

**Figure 8 sensors-23-06840-f008:**
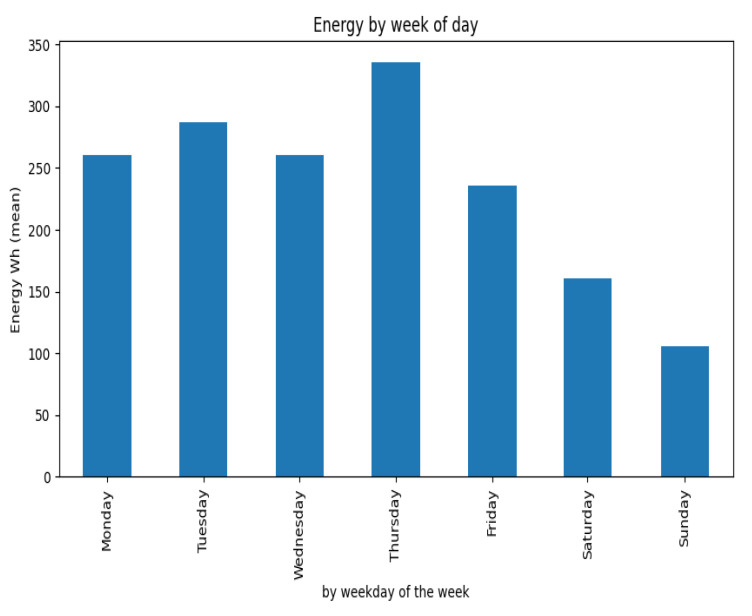
Average room consumption per weekday and season.

**Figure 9 sensors-23-06840-f009:**
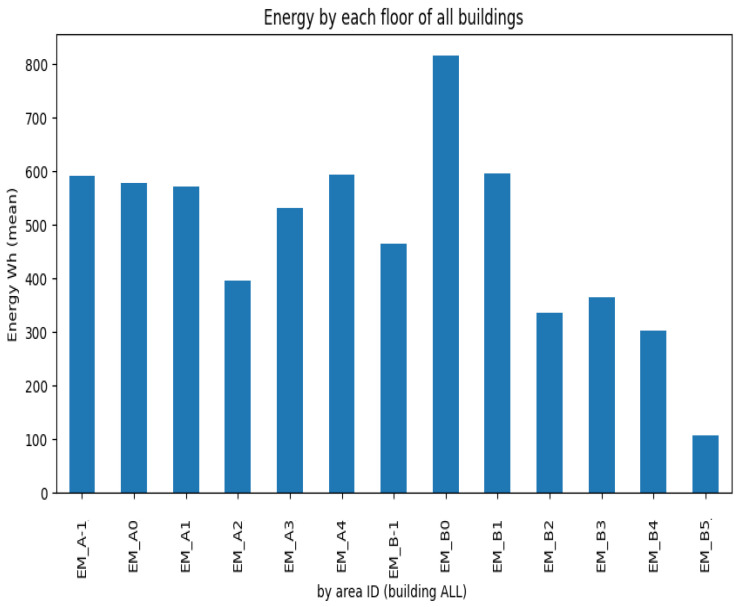
Average consumption per building floor (*A* or *B* means the building block and the number of the floor).

**Figure 10 sensors-23-06840-f010:**
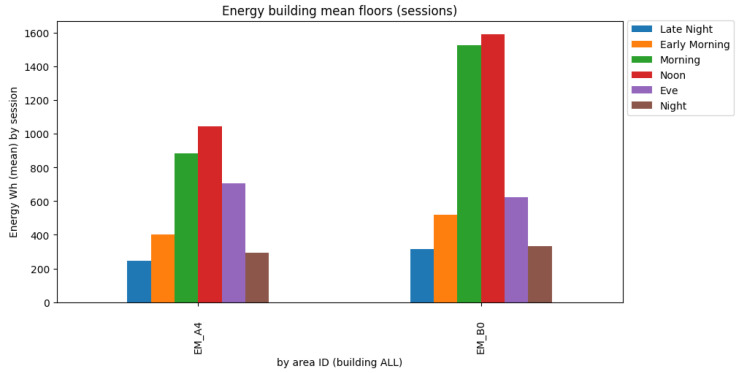
Average session consumption per building floor compared to lower floor consumption.

**Figure 11 sensors-23-06840-f011:**
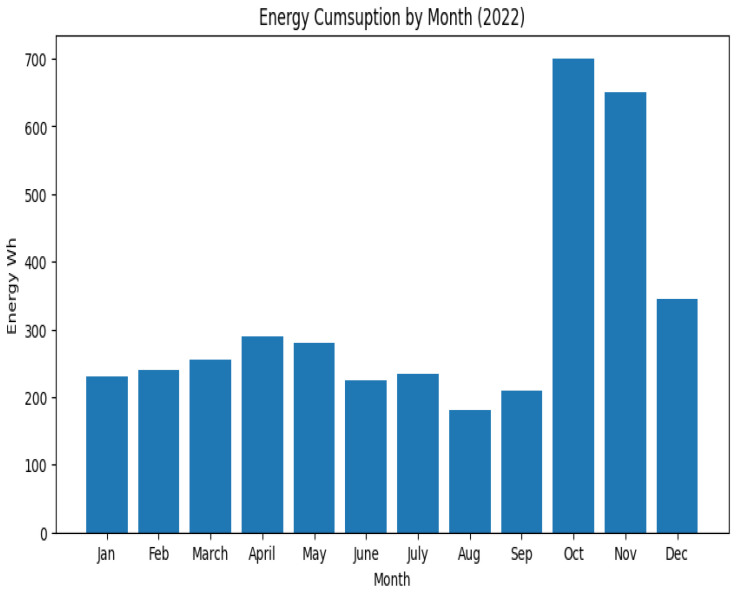
Energy consumption (mean) by month—the year 2022.

**Figure 12 sensors-23-06840-f012:**
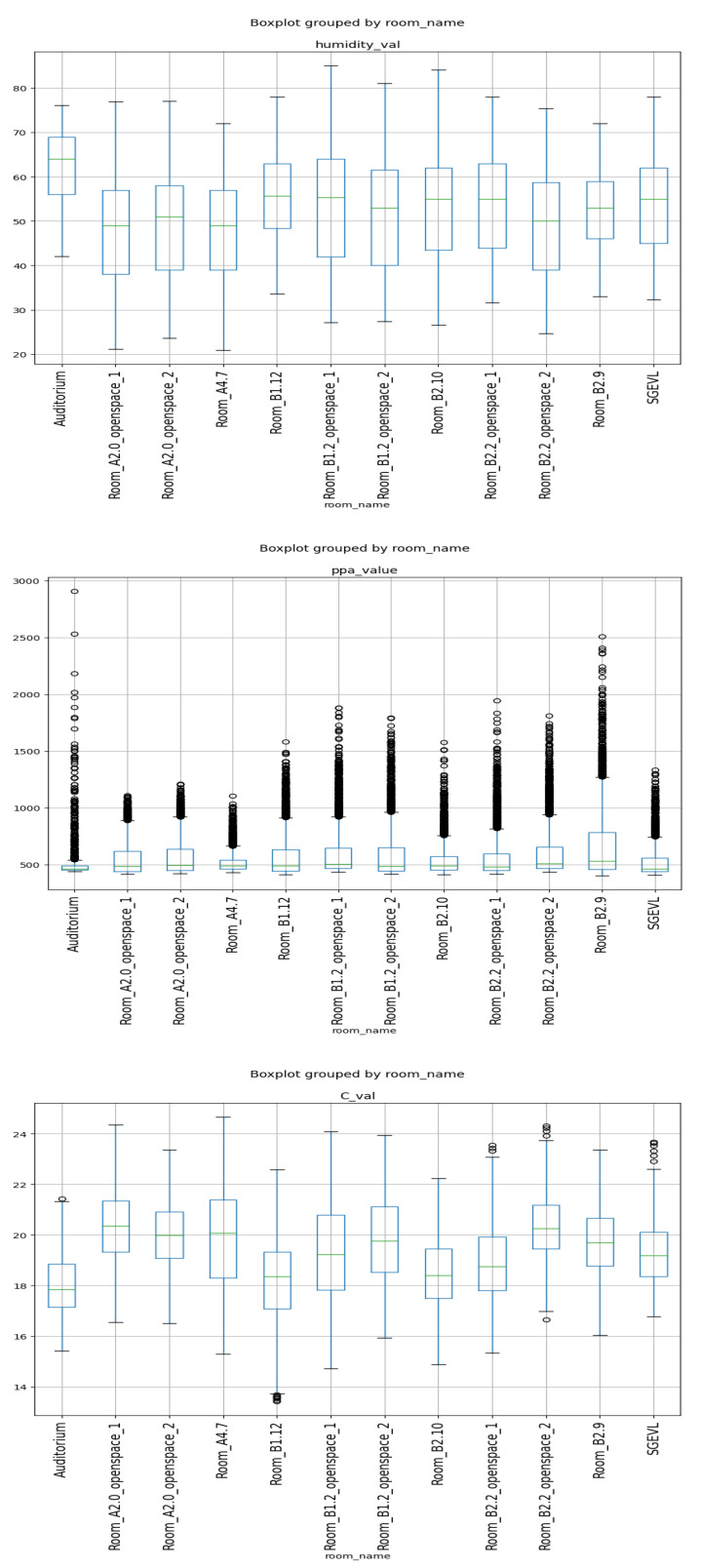
Box-plot of ambient variables by room.

**Figure 13 sensors-23-06840-f013:**
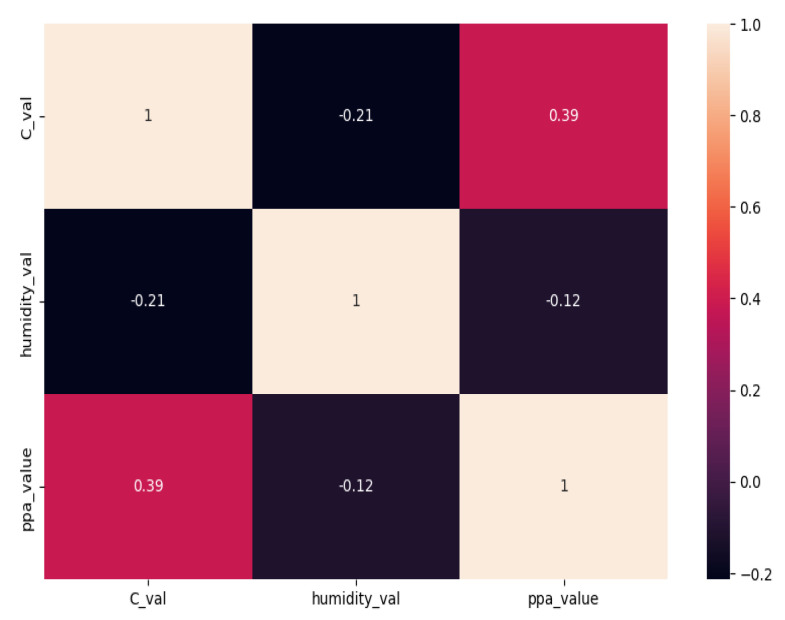
Variable correlation.

**Figure 14 sensors-23-06840-f014:**
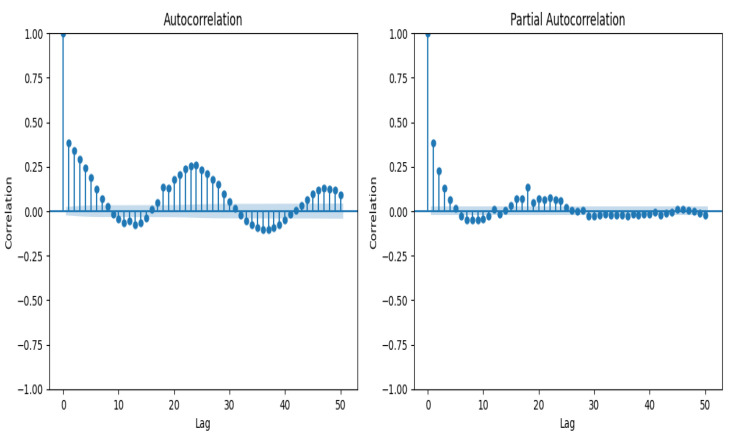
ACF and PACF.

**Figure 15 sensors-23-06840-f015:**
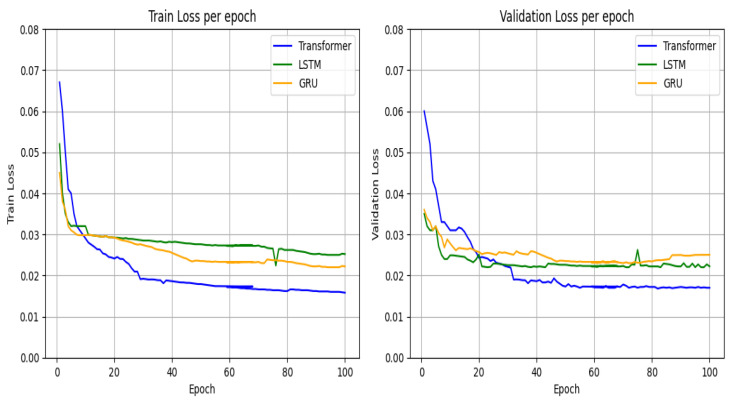
Convergence curves of all evaluate models (training and validation).

**Figure 16 sensors-23-06840-f016:**
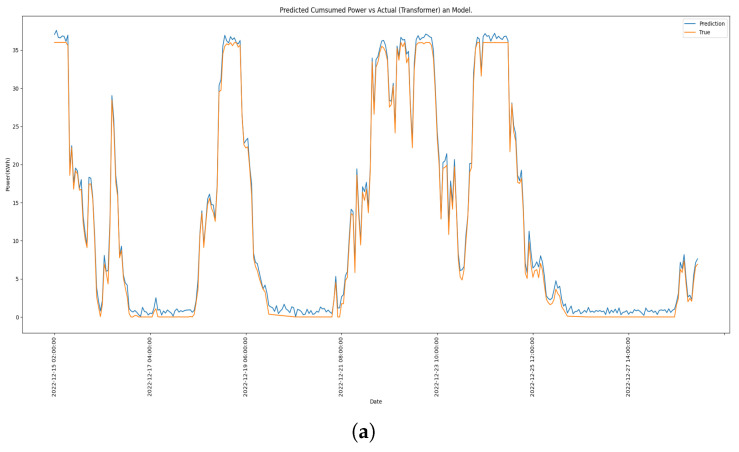
The forecast of energy consumption for each best-performing model. (**a**) Transformer. (**b**) LSTM. (**c**) GRU.

**Table 1 sensors-23-06840-t001:** Results on the models variations (normalized).

LSTM					
**N Cells**	**N Nodes**	**Window**	**Parameters**	**MAPE**	**MSE**
400	128	32	18M	18.11%	15.43%
300	128	32	16M	16.34%	13.35%
200	256	32	13M	17.41%	14.45%
300	256	32	14M	14.26%	11.65%
200	128	24	12M	12.42%	10.02%
250	256	32	14M	**10.02**%	7.04%
**GRU**					
**N Cells**	**N Nodes**	**Window**	**Parameters**	**MAPE**	**MSE**
400	128	32	14M	23.56%	21.43%
300	128	32	13M	15.98%	12.54%
200	256	32	12M	13.93%	11.56%
300	256	12	13M	15.34%	12.76%
200	128	26	11M,	**11.66**%	09.43%
250	128	32	12M	13.59%	10.49%
**Transformer**					
**Heads**	**Enc/Deco**	**Window**	**Parameters**	**MAPE**	**MSE**
10	10/10	32	24M	12.33%	10.61%
10	6/6	32	14M	11.25%	9.75%
10	5/5	32	13M	10.43%	8.27%
6	10/10	32	20M	10.24%	8.11%
6	6/6	32	16M	**7.09**%	5.42%
5	5/5	32	13M	8.36%	6.62%

## Data Availability

Not applicable.

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
