# Peer review of "Transformers for Energy Forecast"

_sensors, 2023, doi:10.3390/s23156840_

Round 1

Reviewer 1 Report

While I concur that the proposed model can be used to forecast energy consumption patterns emanating from DT produced data, this work did not conclusively demonstrate that data used resulted from a DT model. The authors did indicate that live data used were obtained from the Institute for Systems and Computer Engineering, Technology and Science (INESC TEC). It left me wonder why the overemphasis in DT.

With the emphasis in DT, the paper creates an expectation to demonstrate how the DT version of the system (building) was created and implemented as well as how data were obtained using this.

The paper could be better of without this overemphasis in DT models.

The english language used in this article is reasonably good dspite a few typos and misconstructed sentences. The authors should avoid the use of We. Stick to the third person. 

Author Response

Reviewer 1

1)

While I concur that the proposed model can be used to forecast energy consumption patterns emanating from DT produced data, this work did not conclusively demonstrate that data used resulted from a DT model. The authors did indicate that live data used were obtained from the Institute for Systems and Computer Engineering, Technology and Science (INESC TEC). It left me wondering why the overemphasis in DT.

Response:

The main purpose of the developed forecast model is to assess the decision-making process of the building later. This will be integrated with more available information, such as user card entry data and local logins. Energy forecast(consumption can be a proxy to energy efficiency). We want a reasonably capable model to perform multivariate multistep energy forecasting easily. The data was collected from this facility. We reduced the overemphasis on DT, focusing on time series forecasting

==

2)

With the emphasis in DT, the paper creates an expectation to demonstrate how the DT version of the system (building) was created and implemented as well as how data were obtained using this.

Response:

We described how data was gathered from building sensor. The data comes from live sensors placed in the facility that collects several variables periodically. 

==

3)

The paper could be better of without this overemphasis in DT models.

Response:

We just reduce the emphasis in the DT models, although this your final objective is to concretise in the following steps.

4)

The English language used in this article is reasonably good despite a few typos and misconstructed sentences. The authors should avoid the use of We. Stick to the third person.

Response:

  We have reviewed the use of we to address the mentioned problems.

==

Reviewer 2 Report

MDPI Sensors Journal (Manuscript ID: sensors-2455133)

Comments to the Author

This paper performs energy forecasting using transformers on multivariable time-series data. The studied subject is useful for the Sensors audience and technical concepts are clear. However, there are several points need to be addressed to improve the quality of the manuscript.

Suggestions to improve the quality of the paper are provided below:

1)     Please provide the open address of the authors' affiliation. On top of that, please indicate the corresponding author and his/her email address.

2)     Comments about the overall structure

·       Please improve the last sentence of the abstract by splitting it into 2 sentences. Currently, it is very long and hard to follow. Also, please remove “-“ in line 18 for “mean-absolute percentage error”.

·       Same comment applies to the first paragraph of the methodology, stating the objective in lines 133-137. “The main objective of this work is to forecast energy consumption and characterize the room’s usage, allowing it to establish a correlation with main conducted activities  and identify unusual patterns, and perform an overall forecast of energy consumption for the next 10-day period (250 hours), in multistep, allowing to assess the impact of some modifications in the energy consumption.” Almost 5 lines state the objective without giving any break. The objective should be clear and concise. Please break your sentences and make them easy to read and follow.

·       Examining the entire structure throughout the manuscript, I have noticed that this is the main pressing concern for many sentences, where they are structured in long lines without any breakpoints. Although I believe the work presented is important, reading this manuscript is very hard in its current version and several sentences need to be rewritten to improve the clarity.

I highly recommend authors carefully examine and improve these sentence structures throughout the manuscript before resubmitting it. Please utilize proofreading services or obtain the help of a native English speaker to fix this issue to make the manuscript easier for readers to follow and understand.

3)     Please state the novelty of this study together with the contributions in bulletpoints before the outline paragraph in Introduction.

4)     In line 62, the main objective sentence states that “The main objective is to forecast the energy consumption in 10 days, encompassing 1 week and a half of building usage, allowing the application of effective mitigation procedures regarding energy waste and quickly assessing how these mitigation procedures impact the energy true consumption.” This sentence is not clear for me. “forecast the energy consumption in 10 days, encompassing 1 week and a half of building usage” What is half of building usage? How does this objective allow effective mitigation procedures? Please address the clarity of the paragraph.

5)     In Section 3.1 on the topic of LSTM and GRU, please include a few examples of past applications that used these model architectures for time series prediction. For instance, [1] has evaluated the performance of LSTM, Bi-LSTM, GRU, and Bi-GRU to perform occupancy prediction in buildings with different space types. Another example that is related to intelligent vehicle systems can be found in [2], where the authors evaluated the performance of different architectures, including LSTM, to predict vehicle stop activity based on vehicular time-series data. Please refer to the following studies as a good starting point and provide the literature to support the wider use of these algorithms with examples.

[1] Tekler, Z. D., & Chong, A. (2022). Occupancy prediction using deep learning approaches across multiple space types: A minimum sensing strategy. Building and Environment, 226, 109689.

[2] Low, R., Cheah, L., & You, L. (2020). Commercial vehicle activity prediction with imbalanced class distribution using a hybrid sampling and gradient boosting approach. IEEE Transactions on Intelligent Transportation Systems22(3), 1401-1410.

6)     Similarly in Section 3.2, please provide the respective reference on the transformer model architecture, if it is based on previous works to recognise the original implementation.

7)     In the last paragraph of Section 5, line 376 onwards,  the authors mentioned that on top of time-series forecasting, the implementation of transformer models may have many beneficial application areas such as building maintenance. Within the buildings domain, not only building maintenance but also they could be applied to predictive control for HVAC systems [1] which has been a very popular application area and should be highlighted with supporting reference. This information could be either included in this paragraph or for future directions/recommendations.

[1] https://doi.org/10.1016/j.apenergy.2023.12093

Also please provide supporting literature for “building maintenance” in the same paragraph, similar to predictive control. In scientific papers, it is essential to support these claims with literature.

8)     Lastly, in the Conclusion section, the authors mentioned the main limitation of the proposed approach, transformer models, is the need to train the model on richer datasets due to the limited set of training features. While this is absolutely right, I would suggest authors to further carry this discussion by mentioning the availability of public datasets where they could be used to address this problem as future recommendations. Providing a few dataset suggestions to readers would be very beneficial and make the proposed approach practical. For example, ROBOD dataset [1] contains high-resolution building operations and occupancy data to support various applications including occupancy prediction, building simulation and control, energy forecasting and building analytics. Another one is the Building Data Genome Project 2 [2] which contains the energy metering data for 1,636 non-residential buildings. Finally, Global Occupant Behaviour database [3] provides a large compilation of different survey-based and in-situ-based datasets collected from multiple countries. Please refer to these publicly available datasets as practical recommendations and mention these in the manuscript to further strengthen the applicability of proposed the approach.

[1] https://doi.org/10.1007/s12273-022-0925-9

[2] https://doi.org/10.1038/s41597-020-00712-x

[3] https://doi.org/10.1038/s41597-022-01475-3

9)     Minor Issues,

-        In line 69, the last paragraph of the Introduction, please include “Section 6” for the conclusion section.

-        In line 67, missing space between “similarDT”.

-        The text for Figure 8 is too small and the figures seem to be squashed together.

As mentioned in the review report, the quality of the English, more importantly, sentence structures need to be improved. Sentences are generally very long and hard to follow. Please improve them throughout the manuscript based on the comments provided.

Author Response

Reviewer 2

This paper performs energy forecasting using transformers on multivariable time-series data. The studied subject is useful for the Sensors audience and technical concepts are clear. However, there are several points need to be addressed to improve the quality of the manuscript.

Suggestions to improve the quality of the paper are provided below:

1)     Please provide the open address of the authors' affiliation. On top of that, please indicate the corresponding author and his/her email address.

Response:

The template used was not the final, so the mentioned information was missing, in the revised version the information is now included in the correct version and the journal texstyle.

==

2)     Comments about the overall structure

  •       Please improve the last sentence of the abstract by splitting it into 2 sentences. Currently, it is very long and hard to follow. Also, please remove “-“ in line 18 for “mean-absolute percentage error”.

  •       Same comment applies to the first paragraph of the methodology, stating the objective in lines 133-137. “The main objective of this work is to forecast energy consumption and characterise the room’s usage, allowing it to establish a correlation with main conducted activities  and identify unusual patterns, and perform an overall forecast of energy consumption for the next 10-day period (250 hours), in multistep, allowing to assess the impact of some modifications in the energy consumption.” Almost 5 lines state the objective without giving any break. The objective should be clear and concise. Please break your sentences and make them easy to read and follow.

  •       Examining the entire structure throughout the manuscript, I have noticed that this is the main pressing concern for many sentences, where they are structured in long lines without any breakpoints. Although I believe the work presented is important, reading this manuscript is very hard in its current version and several sentences need to be rewritten to improve the clarity.

I highly recommend authors carefully examine and improve these sentence structures throughout the manuscript before resubmitting it. Please utilize proofreading services or obtain the help of a native English speaker to fix this issue to make the manuscript easier for readers to follow and understand.

Response:

We revised the manuscript, simplified the main objective and reduce the sentence density.

==

3)     Please state the novelty of this study together with the contributions in bulletpoints before the outline paragraph in Introduction.

 Response:

we added the following sentence: The energy forecast model is based on modifying the multi-attention transformer model by including a learnable weighting feature attention matrix to address the building energy efficiency.

==

4)     In line 62, the main objective sentence states that “The main objective is to forecast the energy consumption in 10 days, encompassing 1 week and a half of building usage, allowing the application of effective mitigation procedures regarding energy waste and quickly assessing how these mitigation procedures impact the energy true consumption.” This sentence is not clear for me. “forecast the energy consumption in 10 days, encompassing 1 week and a half of building usage” What is half of building usage? How does this objective allow effective mitigation procedures? Please address the clarity of the paragraph.

Response:

This was a wrong sentence, we changed to: forecast the energy consumption of next 10 days.

==

5)     In Section 3.1 on the topic of LSTM and GRU, please include a few examples of past applications that used these model architectures for time series prediction. For instance, [1] has evaluated the performance of LSTM, Bi-LSTM, GRU, and Bi-GRU to perform occupancy prediction in buildings with different space types. Another example that is related to intelligent vehicle systems can be found in [2], where the authors evaluated the performance of different architectures, including LSTM, to predict vehicle stop activity based on vehicular time-series data. Please refer to the following studies as a good starting point and provide the literature to support the wider use of these algorithms with examples.

[1] Tekler, Z. D., & Chong, A. (2022). Occupancy prediction using deep learning approaches across multiple space types: A minimum sensing strategy. Building and Environment, 226, 109689.

[2] Low, R., Cheah, L., & You, L. (2020). Commercial vehicle activity prediction with imbalanced class distribution using a hybrid sampling and gradient boosting approach. IEEE Transactions on Intelligent Transportation Systems, 22(3), 1401-1410.

6)     Similarly in Section 3.2, please provide the respective reference on the transformer model architecture, if it is based on previous works to recognise the original implementation.

Response: 

We base the work on the well know work of vaswani 2017, however we modified the way inputs are combined into the embedding and employed a variation of the single-attention head into multi-attention heads. We reference the base work accordingly. The mentioned works we have included in the literature section are relevant to the work.

==

7)     In the last paragraph of Section 5, line 376 onwards,  the authors mentioned that on top of time-series forecasting, the implementation of transformer models may have many beneficial application areas such as building maintenance. Within the buildings domain, not only building maintenance but also they could be applied to predictive control for HVAC systems [1] which has been a very popular application area and should be highlighted with supporting reference. This information could be either included in this paragraph or for future directions/recommendations.

[1] https://doi.org/10.1016/j.apenergy.2023.12093

Also please provide supporting literature for “building maintenance” in the same paragraph, similar to predictive control. In scientific papers, it is essential to support these claims with literature.

Response:

Also The mentioned references actually has some error and we cannot see if its suitable, however we followed the proposal and added an extra one regarding the building subject and predictive maintenance. 

==

8)     Lastly, in the Conclusion section, the authors mentioned the main limitation of the proposed approach, transformer models, is the need to train the model on richer datasets due to the limited set of training features. While this is absolutely right, I would suggest authors to further carry this discussion by mentioning the availability of public datasets where they could be used to address this problem as future recommendations. Providing a few dataset suggestions to readers would be very beneficial and make the proposed approach practical. For example, ROBOD dataset [1] contains high-resolution building operations and occupancy data to support various applications including occupancy prediction, building simulation and control, energy forecasting and building analytics. Another one is the Building Data Genome Project 2 [2] which contains the energy metering data for 1,636 non-residential buildings. Finally, Global Occupant Behaviour database [3] provides a large compilation of different survey-based and in-situ-based datasets collected from multiple countries. Please refer to these publicly available datasets as practical recommendations and mention these in the manuscript to further strengthen the applicability of proposed the approach.

[1] https://doi.org/10.1007/s12273-022-0925-9

[2] https://doi.org/10.1038/s41597-020-00712-x

[3] https://doi.org/10.1038/s41597-022-01475-3

Response:

 We were aware of some of the datasets, namely the ROBOD and Building data Genome,  however iROBOD only encompasses a total of 181 days, The other two we have included in text and will be subject for further studies..

==

9)     Minor Issues,

-        In line 69, the last paragraph of the Introduction, please include “Section 6” for the conclusion section.

-        In line 67, missing space between “similarDT”.

-        The text for Figure 8 is too small and the figures seem to be squashed together.

 Response:

We modified the figures and set a different way of presenting.

== 

Comments on the Quality of English Language

As mentioned in the review report, the quality of the English, more importantly, sentence structures need to be improved. Sentences are generally very long and hard to follow. Please improve them throughout the manuscript based on the comments provided.

Response:

The comments were addressed and were relevant to improve paper quality and readability.

Reviewer 3 Report

The manuscript sensors-2455133 presents a modified transformer model focused on multi-variable time series used to predict the energy consumption of buildings. Several improvements are necessary for publication, regarding the structure of the manuscript, method and experiment description, and English language. I have the following observations:

1. The original contributions might be emphasized in the introduction.

2. Please reformulate: "Fault prediction in buildings using DT subject of study by [8] [...]".

3. Please revise: "[...] such as the example, the shutdown of production lines, or inspection of faulty turbines".

4. Why energy is written in some places with capital first letter?

5. The literature review is quite short for such a journal paper. Some existing electricity consumption forecasting methods are not mentioned (e.g. ARIMA, TBATS, fuzzy logic, etc.).

6. The description of LSTM and GRU is not well placed in Section 3. LSTM and GRU are not part of the proposed method, these models being used only for comparisons. Subsection 3.1. should be shortened and moved into Section 2.

7. Please revise: "[...] to maps the input sequence [...]".

8. Please revise: "an numeric token" (on line 178).

9. Please revise: "Considering a embedding vector [...]".

10. Figure 4 needs a better explanation.

11. Figure 6 needs an explanation.

12. please revise: "cafe machines".

13. The sentence from line 299 ends with a comma.

14. The graphics from Figure 12 are too small. Maybe it is better to place them in different larger figures.

15. Please revise: "To accurate forecast energy consumption [...]" on line 305.

16. Please revise: "Furthermore, while simultaneously compared the performance with LSTM and GRU-based models."

17. Please check: "Regarding LSTM the optimal number of nodes is 250, 256 nodes per layer".

18. Please check: "Regarding GRU, the optimal number of nodes is 200, 128 nodes per layer".

19. In all the three graphics from Figure 16, there is missing from both axes what are they representing. Please, introduce all that information.

20. Table 1 presents MAPE, but does not present MSE. Then why MSE is given in subsection 4.2? And why Section 6 claims that "MSE and MAPE were used to evaluate the models"? That seems to be not true. Please solve such contradictions in the manuscript.

21. Please revise the sentence (in Section 6): "The results the performance of the multivariate transformer model using multi-head attention mechanism, allowing to improve of the MAPE value by almost 3.2% over the best-trained baselines."

22. It is not clear how the proposed forecasting method is integrated into the DT. The title of the manuscript, the Abstract, the Introduction, as well as the Literature Review, put a high focus on DT. But sections 3,4 and 5, which constitute the main part of the manuscript (in fact the contribution of the authors) mention just marginally the DT. Please, decide what are you focusing on and adjust the paper correspondingly.

The manuscript might be improved from the English language point of view.

Author Response

Reviewer 3

Comments and Suggestions for Author

The manuscript sensors-2455133 presents a modified transformer model focused on multi-variable time series used to predict the energy consumption of buildings. Several improvements are necessary for publication, regarding the structure of the manuscript, method and experiment description, and English language. I have the following observations:

  1. The original contributions might be emphasized in the introduction.

Response:

We rephrase this to focus on the modifications that were introduced in the transformer model

== 

  1. Please reformulate: "Fault prediction in buildings using DT subject of study by [8] [...]".

Response:

We rephrased the sentence,

== 

  1. Please revise: "[...] such as the example, the shutdown of production lines, or inspection of faulty turbines".

Response:

We rephrased the sentence,

== 

  1. Why energy is written in some places with capital first letter?

Response:

These were typos, thanks for noticing

== 

  1. The literature review is quite short for such a journal paper. Some existing electricity consumption forecasting methods are not mentioned (e.g. ARIMA, TBATS, fuzzy logic, etc.).

Response:

The mentioned models(arima Tbats)  are not well suited for multivariate time series; They are targeted for univariate. The work presented explores the use of multivariate multistep time series forecasting based models, 

== 

  1. The description of LSTM and GRU is not well placed in Section 3. LSTM and GRU are not part of the proposed method. These models being used only for comparisons. Subsection 3.1. should be shortened and moved into Section 2.

Response:

We agree, several works also employ the use of same models with similar purposes, we just collapsed into a single subsection more reduced

== 

  1. Please revise: "[...] to maps the input sequence [...]".

We rephrased the sentence,

== 

  1. Please revise: "an numeric token" (on line 178).

We rephrased the sentence,

== 

  1. Please revise: "Considering a embedding vector [...]".

We rephrased the sentence,

== 

  1. Figure 4 needs a better explanation.

We add the description in the text of the process represented.

  1. Figure 6 needs an explanation.

more description and reformulation was made

  1. please revise: "cafe machines".

check

== 

  1. The sentence from line 299 ends with a comma.
  2. The graphics from Figure 12 are too small. Maybe it is better to place them in different larger figures.

checked, we put in three different images

  1. Please revise: "To accurate forecast energy consumption [...]" on line 305.

Check

  1. Please revise: "Furthermore, while simultaneously comparing the performance with LSTM and GRU-based models."

Check

  1. Please check: "Regarding LSTM the optimal number of nodes is 250, 256 nodes per layer".

check

  1. Please check: "Regarding GRU, the optimal number of nodes is 200, 128 nodes per layer".

check

  1. In all three graphics from Figure 16, there is missing from both axes what they are representing. Please, introduce all that information.

information was added to figure and caption

  1. Table 1 presents MAPE but does not present MSE. Then why is MSE given in subsection 4.2? And why Section 6 claims that "MSE and MAPE were used to evaluate the models"? That is not true. Please solve such contradictions in the manuscript.

We guided the training using MSE, but due to latex error, the mse was not appearing in the table; now its included.

  1. Please revise the sentence (in Section 6): "The results the performance of the multivariate transformer model using multi-head attention mechanism, allowing to improve of the MAPE value by almost 3.2% over the best-trained baselines."

check

  1. It is not clear how the proposed forecasting method is integrated into the DT. The title of the manuscript, the Abstract, the Introduction, and the Literature Review put a high focus on DT. But sections 3,4 and 5, which constitute the main part of the manuscript (in fact, the contribution of the authors) mention just marginally the DT. Please, decide what you are focusing on and adjust the paper correspondingly.

We have reduced the overemphasis on dt, and focusing more on the multivariate time series energy forecast.

The manuscript might be improved from the English language point of view.

Reviewer 4 Report

The following suggestions recommended to this article are,

1. What is the predicted quantitative value using this proposed approach?. It would be added in abstract to improve the readability. 

2. The authors mentioned in abstract that "modified transformer model used". Can you highlight the challenges faced in the existing transformer ?

3. Suggested to use mathematical formula to derive input and output based on Long Short-term Memory and Gate Recurrent Unit (LSTM and GRU). 

4. Equations are not cited in context. It should be cited in appropriate lines. 

5. In Figure 5, what is the significance of multi head attention? What are the parameters considered for paying attention. 

6. This work is based on digital twin model. It should show the digital replica of original and describe how this model works. This part completely missing to define the proposed work. 

7. Figure 2. LSTM cell block- here, intermediate lost has not calculated/ included. Have train the model without loss?

8. Figure 3. GRU cell block- Are you giving equal number of feedback or different number of inputs?. It should describe below the figure. 

9. In section 4, complete data source are not mentioned but results graphs are plotted. Can you show where the data samples are available?

10. In Figure 14. ACF and PACF- What is the unit value of Y axis?

11. Figure 15,16- Show the x and Y axis values?

Focus on Section 2 and 3 to improve the english

Author Response

Reviewer 4 

Comments and Suggestions for Authors

The following suggestions recommended for this article are,

  1. What is the predicted quantitative value using this proposed approach?. It would be added in the abstract to improve the readability. 

Response: 

-We want to forecast the energy consumption given a set o input variables and past consumption history data. We added this information to the revised version

  1. The authors mentioned in the abstract that a "modified transformer model was used". Can you highlight the challenges faced in the existing transformer?

Response: 

The conventional transformer does not model multivariate inputs in a straight manner. For this, we modified the input of the encoder transformer to incorporate multi-feature input through the combination of equation 6, which adds an extra learnable attention matrix as a learnable matrix weight of the feature inputs towards the objective in combination with the multi-head attention variation.

  1. Suggested using a mathematical formula to derive input and output based on Long Short-term Memory and Gate Recurrent Unit (LSTM and GRU). 

Response: 

added

  1. Equations are not cited in context. It should be cited in appropriate lines. 

Response: 

We now have cited the mentioned equations

  1. In Figure 5, what is the significance of multi-head attention? What are the parameters considered for paying attention? 

Response: 

The multi-head attention means that for a given position (token), they can be contributions from several attention heads. The relevant parameters are the number of attention heads itself. While this makes the model more suitable to incorporate more relevant information, it can lead to overfitting and an increase in model complexity in an exponential way.

  1. This work is based on a digital twin model. It should show the digital replica of the original and describe how this model works. This part completely missing to define the proposed work. 

Response: 

The objective is to feed with the given input data from the past 250 instants and forecast the energy question of the next 250 steps. 

  1. Figure 2. LSTM cell block- here, the intermediate loss has not been calculated/ included. Have you trained the model without loss?

Response: 

All models are guided by the same loss metric. Actually, figure 2 only represents the lstm block. 

  1. Figure 3. GRU cell block- Are you giving an equal number of feedback or a different number of inputs?. It should describe in the figure. 

Response: 

The Gru block is a variation of LSTM with a shorter number of parameters. The number of inputs is the same on all models. This figure only corresponds to the GRU block itself. A more detailed description is in section 5 where the model's scheme with different inputs is presented (a thought in a generalised way).

  1. In section 4, complete data source are not mentioned but results graphs are plotted. Can you show where the data samples are available?

Response: 

We removed the picture of the full data,  since the series is was to long to be represented in a single image. The images mentioned refer to the forecast of energy consumption vs true consumption in the test set. We add a sample of the data.

  1. In Figure 14. ACF and PACF- What is the unit value of Y axis?

Response: 

Is the correlation value [-1 to 1] as expressed in the plot for both figures? Actually, x is lag and not epoch as stated. 

  1. Figure 15,16- Show the x and Y axis values?

Response: 

Todo x-axis, timesteps, y energy in Wh,  mentioned now in the plots.

Focus on Section 2 and 3 to improve the English

done

Round 2

Reviewer 1 Report

Minor edits are still required.

Minor edits are still required.

Author Response

We had some modifications to the text in response to the reviewer's questions

Reviewer 2 Report

Thank you for addressing my comments and revising the manuscript accordingly. 

I have a minor comment regarding the public datasets suggested for model training in future works. The authors mention that one of the datasets, ROBOD [https://doi.org/10.1007/s12273-022-0925-9], has only 181 days of data and is therefore excluded. However, I believe that this dataset is suitable for many energy-related applications as it contains a multitude of features (such as indoor and outdoor environmental conditions, Wi-Fi, energy consumption of end uses (i.e., HVAC, lighting, plug loads and fans), HVAC operations together with occupancy) within a building. In fact, it is more comprehensive than many datasets within the Global Occupant Behavior Database. The variety of features and the suitability of the chosen dataset are as important as the size of the data for model training. Therefore, I highly recommend mentioning this dataset alongside the other two datasets in this section.

After addressing this comment, this manuscript will be ready for publication. Overall, great job!

The level of English is satisfactory.

Author Response

Reviewer 02:

Thank you for addressing my comments and revising the manuscript accordingly. 

I have a minor comment regarding the public datasets suggested for model training in future works. The authors mention that one of the datasets, ROBOD [https://doi.org/10.1007/s12273-022-0925-9], has only 181 days of data and is therefore excluded. However, I believe that this dataset is suitable for many energy-related applications as it contains a multitude of features (such as indoor and outdoor environmental conditions, Wi-Fi, energy consumption of end uses (i.e., HVAC, lighting, plug loads and fans), HVAC operations together with occupancy) within a building. In fact, it is more comprehensive than many datasets within the Global Occupant Behavior Database. The variety of features and the suitability of the chosen dataset are as important as the size of the data for model training. Therefore, I highly recommend mentioning this dataset alongside the other two datasets in this section.

After addressing this comment, this manuscript will be ready for publication. Overall, great job!

Response

We added the following paragraph and referred the work in the document:

The construction of a richer dataset must follow the guidelines of the ROBOD project \cite{tekler2022robod}, which encompasses many sensors, HVAC, building occupancy, and wifi traffic, among others. 

The ROBOD dataset is a broader holistic dataset indeed, and we have found guidelines that enable us to construct the dataset similarly. The data is being collected following ROBOD standards ideas. it was a great help.

Reviewer 3 Report

The manuscript sensors-2455133 was significantly improved but some of the problems are still not solved:

1. In all three graphics from Figure 16, there is still missing from both axes what they are representing. Please, introduce all that information into the figures. It is far better to have this information directly in the charts and not in the caption. It is unusual to give it in the caption.

2. It is not clear how the proposed forecasting method is integrated into the DT. The title of the manuscript, the Abstract, the Introduction, and the Literature Review put a high focus on DT. But sections 3, 4 and 5, which constitute the main part of the manuscript (in fact, the contribution of the authors) have nothing about the DT. Please, decide what you are focusing on and adjust the paper correspondingly. If you decide to not present how the forecasting method is integrated into the DT, please avoid mentioning it in the title, and remove it from the Abstract, Introduction, and Literature Review. It is too much focus on DT in those sections.

The English was improved.

Author Response

Reviewer 03

The manuscript sensors-2455133 was significantly improved but some of the problems are still not solved:

  1. In all three graphics from Figure 16, there is still missing from both axes what they are representing. Please, introduce all that information into the figures. It is far better to have this information directly in the charts and not in the caption. It is unusual to give it in the caption.

Response:

We have redone the plots and the axis are now represented the axis, dates of forecast accordly

  1. It is not clear how the proposed forecasting method is integrated into the DT. The title of the manuscript, the Abstract, the Introduction, and the Literature Review put a high focus on DT. But sections 3, 4 and 5, which constitute the main part of the manuscript (in fact, the contribution of the authors) have nothing about the DT. Please, decide what you are focusing on and adjust the paper correspondingly. If you decide to not present how the forecasting method is integrated into the DT, please avoid mentioning it in the title, and remove it from the Abstract, Introduction, and Literature Review. It is too much focus on DT in those sections.

Response:

We have redone the document and removed the majority of mentions to DT and reframed the sections focusing on the forecasting of energy consumption works and related tasks.

Round 3

Reviewer 3 Report

The manuscript sensors-2455133 was significantly improved, all the problems being solved. I recommend to be accepted for publication. For readability, I suggest to increase the font for all the texts from Fig. 16.